# Quercetin Attenuates Acetaldehyde-Induced Cytotoxicity via the Heme Oxygenase-1-Dependent Antioxidant Mechanism in Hepatocytes

**DOI:** 10.3390/ijms25169038

**Published:** 2024-08-20

**Authors:** Kexin Li, Minori Kidawara, Qiguang Chen, Shintaro Munemasa, Yoshiyuki Murata, Toshiyuki Nakamura, Yoshimasa Nakamura

**Affiliations:** 1Graduate School of Environmental and Life Science, Okayama University, Okayama 700-8530, Japan; lkx0903@hnagri.org.cn (K.L.); pyk49ulx@s.okayama-u.ac.jp (M.K.); smunemasa@okayama-u.ac.jp (S.M.); muta@cc.okayama-u.ac.jp (Y.M.); t-nakamura@okayama-u.ac.jp (T.N.); 2School of Food Science and Technology, Dalian Polytechnic University, Dalian 116034, China; p6zw75n8@s.okayama-u.ac.jp; 3Graduate School of Environmental, Life, Natural Science and Technology, Okayama University, Okayama 700-8530, Japan

**Keywords:** quercetin, acetaldehyde, glutathione, aldehyde dehydrogenase, heme oxygenase-1

## Abstract

It is still unclear whether or how quercetin influences the toxic events induced by acetaldehyde in hepatocytes, though quercetin has been reported to mitigate alcohol-induced mouse liver injury. In this study, we evaluated the modulating effect of quercetin on the cytotoxicity induced by acetaldehyde in mouse hepatoma Hepa1c1c7 cells, the frequently used cellular hepatocyte model. The pretreatment with quercetin significantly inhibited the cytotoxicity induced by acetaldehyde. The treatment with quercetin itself had an ability to enhance the total ALDH activity, as well as the ALDH1A1 and ALDH3A1 gene expressions. The acetaldehyde treatment significantly enhanced the intracellular reactive oxygen species (ROS) level, whereas the quercetin pretreatment dose-dependently inhibited it. Accordingly, the treatment with quercetin itself significantly up-regulated the representative intracellular antioxidant-related gene expressions, including heme oxygenase-1 (HO-1), glutamate-cysteine ligase, catalytic subunit (GCLC), and cystine/glutamate exchanger (xCT), that coincided with the enhancement of the total intracellular glutathione (GSH) level. Tin protoporphyrin IX (SNPP), a typical HO-1 inhibitor, restored the quercetin-induced reduction in the intracellular ROS level, whereas buthionine sulphoximine, a representative GSH biosynthesis inhibitor, did not. SNPP also cancelled the quercetin-induced cytoprotection against acetaldehyde. These results suggest that the low-molecular-weight antioxidants produced by the HO-1 enzymatic reaction are mainly attributable to quercetin-induced cytoprotection.

## 1. Introduction

Acetaldehyde, one of the major metabolites of ethanol, is widely regarded as a main mediator of several ethanol-induced abnormal behaviors and chronic diseases, such as carcinogenesis, liver diseases, and neuropsychic disorders [1]. Acetaldehyde directly modifies the amino acid residues of proteins and DNA to form covalent conjugates by its electrophilic reaction [2]. These acetaldehyde adducts might cause protein structure alteration and DNA damage, resulting in the modulation of cellular homeostasis and survival. Aldehyde dehydrogenases (ALDHs) mainly contribute to the aldehyde metabolism [3]. Out of the 19 human ALDHs, mitochondrial ALDH2 is an essential regulator of the acetaldehyde metabolism in the liver, and cytosolic ALDH1A1 additionally supports the acetaldehyde metabolism [3,4]. About half of East Asian people have a polymorphism in the ALDH2 gene, resulting in the continuous accumulation of acetaldehyde, which is likely to elevate the risk of chronic diseases related to alcohol, including cancer [5]. It is thus conceivable that the intake of certain food ingredients enhancing the activities of liver ALDH is a promising strategy to limit acetaldehyde-induced toxic events.

The excessive accumulation of reactive oxygen species (ROS) directly modulates the redox homeostatic balance and modifies biomolecules, both of which are associated with the development of degenerative diseases [6]. ROS are also by-products of ethanol oxidation via the microsomal ethanol oxidizing system (MEOS), which is upregulated by the chronic consumption of ethanol [7]. Furthermore, the imbalance between the production and disposition of acetaldehyde is closely linked to mitochondria dysfunction and thus excessive ROS production [8]. Thus, oxidative stress is profoundly involved in alcohol-related toxic events. The nuclear factor erythroid 2-related factor 2 (Nrf2) takes part in antioxidative cytoprotection against ethanol- or acetaldehyde-induced toxicity [9]. Nrf2 transcriptionally activates the gene expression of various antioxidative/cytoprotective proteins including heme oxygenase-1 (HO-1) and quinone oxidoreductase 1 (NAD(P)H dehydrogenase [quinone] 1; NQO1). HO-1 produces two low-molecular-weight antioxidants, carbon oxide (CO) and biliverdin, as well as a ferrous ion during the catalytic decomposition of heme [10]. HO-1 is also involved in cytoprotection independent of the oxygenase enzymatic activity or reaction products and possibly through the alteration of the nuclear transcriptional factor activity [11]. Actually, sulforaphane, a well-known Nrf2-activating organosulfur compound, suppresses mycoplasma-induced pathological inflammatory injury though the HO-1 enzyme activity-dependent mechanism [12]. Thus, HO-1 has been regarded as one of the representative biomarkers for screening biologically active antioxidants [13].

Quercetin is the most famous flavonoid aglycone due to its human-health-maintaining potential [14]. Quercetin shows not only free radical scavenging activities, but also modulating effects on the damage of lipids, proteins and DNA [15]. Furthermore, quercetin has the potential to modulate cellular signal pathways, including the Nrf2 pathway, resulting in the upregulation of antioxidative gene expression [16]. Quercetin has been considered as a promising dietary factor for the prevention of various life-style-related diseases related to ROS-mediated toxic events [14,15]. In addition, quercetin was found to attenuate the liver injury induced by excessive alcohol intake in a mouse model [17]. Although quercetin has the potential to improve alcohol-related diseases, it remains to be clarified whether or how quercetin influences the toxic events induced by acetaldehyde in hepatocytes. Furthermore, the involvement of the HO-1-dependent antioxidant mechanism in the protection of cells against acetaldehyde by quercetin remains to be clarified, even though it is well known that quercetin activates the intracellular antioxidative defense system [15].

In this study, quercetin was evaluated as a potential cytoprotector against acetaldehyde-induced cytotoxicity in the cultured hepatocyte model. We also demonstrated that quercetin significantly enhanced the gene expression of intracellular antioxidant-related proteins as well as the main ALDH isozymes for acetaldehyde oxidation. Furthermore, the ROS accumulation, as well as cytotoxicity induced by acetaldehyde, was reduced by quercetin in a HO-1-dependent manner.

## 2. Results

### 2.1. Cytoprotection against the Acetaldehyde Toxicity and ALDH Activity Enhancement by Quercetin

We first tested whether quercetin affected the toxicity induced by acetaldehyde in the cultured hepatocyte model, Hepa1c1c7 cells. After the 24 h pretreatment with quercetin, acetaldehyde (10 mM) was treated with the cells for 3 h. The treatment of acetaldehyde alone decreased the cell viability to 82% (Figure 1). On the contrary, the quercetin pretreatment dose-dependently attenuated the cytotoxicity induced by acetaldehyde, with the cell viability recovering to more than 90% (Figure 1). The significant effect of quercetin was observed from 5 μM. Next, we investigated whether quercetin modulates the gene expression of acetaldehyde-metabolizing enzymes. ALDH1A1, ALDH2, and ALDH3A1, the conventional ALDHs expressed in hepatocytes, are suggested to contribute to the acetaldehyde metabolism [18,19]. The ALDH1A1 and ALDH3A1 gene expressions were significantly up-regulated by quercetin alone, whereas that of ALDH2, the master enzyme for acetaldehyde metabolism in the liver (Figure 2A), was not. Based on this observation, we next examined whether quercetin influenced the activity of ALDH enzymes. As shown in Figure 2B, the quercetin treatment significantly and concentration-dependently enhanced the total ALDH enzyme activity in Hepa1c1c7 cells (Figure. 2B). Treatment with 5 μM of quercetin increased it about 2-fold compared to that of the control.

### 2.2. Inhibitory Effect of Quercetin on the Reactive Oxygen Species Accumulation Induced by Acetaldehyde

Oxidative stress is regarded as one possible mechanism implicated in acetaldehyde toxic events [20]. Therefore, we evaluated the changes in the intracellular ROS level using the dichlorofluorescin diacetate (DCFH-DA) assay, which is frequently utilized for the measurement of intracellular ROS. Actually, the treatment of acetaldehyde itself (10 mM) significantly enhanced the intracellular ROS level about 1.4-fold compared to the control (Figure 3). However, the quercetin pretreatment dose-dependently attenuated the ROS level enhanced by acetaldehyde (Figure 3).

### 2.3. Up-Regulation of the Intracellular Antioxidant-Related Gene Expression by Quercetin

Since quercetin had the potential to inhibit the oxidative stress as well as cytotoxicity induced by acetaldehyde, we examined whether the antioxidant-related proteins were involved in the underlying mechanism. We focused on the enzymes producing low-molecular-weight antioxidant molecules, glutamate-cysteine ligase, catalytic subunit (GCLC) and cystine/glutamate exchanger (xCT) as well as HO-1 [13]. The treatment with quercetin alone significantly up-regulated the expressions of all the genes (Figure 4A). Quercetin itself significantly increased the expression levels of HO-1, GCLC, and xCT about 1.6-fold, 1.3-fold, and 2.2-fold, respectively, compared to the control. Accordingly, quercetin also significantly elevated the intracellular level of glutathione (GSH) (Figure 4B), as well as the HO-1 protein level (Figure 4C). The significant effect of quercetin was observed from 5 μM. These results suggested that the uptake of cystine as well as the enhanced biosynthesis might involve the increased GSH level.

### 2.4. HO-1-Denpedent Inhibition of the Acetaldehyde-Induced ROS Elevation and Cytotoxicity by Quercetin

Since the gene expressions of both HO-1 and the GSH biosynthesis-related proteins were enhanced by quercetin, experiments using tin protoporphyrin IX (SNPP), a typical inhibitor of HO-1, as well as buthionine sulphoximine (BSO), a representative inhibitor of GSH biosynthesis, were performed. The concentrations of the used inhibitors were based on previous reports showing significant inhibition without cytotoxicity [21,22]. SNPP significantly cancelled the attenuation of the ROS level by quercetin (Figure 5A). On the other hand, BSO did not significantly affect it (Figure 5B). Furthermore, SNPP at the same concentration completely abolished the quercetin-induced cytoprotection against acetaldehyde in Hepa1c1c7 cells (Figure 6). Taken together, the cytoprotective mechanism of quercetin might involve the production of low-molecular-weight antioxidants dependent on HO-1 rather than GSH biosynthesis.

## 3. Discussion

In the present study, we demonstrated the cytoprotective potential of quercetin against the toxic events induced by acetaldehyde using the cultured hepatocyte model (Figure 1). We also suggested that quercetin possesses the potential to enhance the total ALDH activity (Figure 2B), even though quercetin has been reported to inhibit the ALDH enzyme activity in in vitro experiments [23]. The effective concentration of quercetin required to enhance the ALDH activity (5 μM) was equivalent to that of 3,4-dihydroxyphenylacetic acid, 3-hydroxyphenylacetic acid and benzyl isothiocyanate [24,25,26], suggesting that quercetin is also a potential inducer for the enhancement of the acetaldehyde metabolism. We have very recently reported that quercetin also shows cytoprotective effects against acetaldehyde in human hepatocellular carcinoma HepG2 cells [27], suggesting that quercetin is a potential cytoprotector against acetaldehyde not only in mice, but also in humans. Quercetin significantly up-regulated the expression of ALDH1A1 and ALDH3A1 genes, the ALDH2-assisting enzymes for the acetaldehyde metabolism [18,19], whereas it did not change that of ALDH2 (Figure 2A). The transcriptional regulation of ALDH1A1 or ALDH3A1 might be involved in the total quercetin-enhanced activity of ALDH. Our recent study demonstrated that an intestinal metabolite of quercetin protected the cells from the cytotoxicity induced by acetaldehyde, dependent on the enhancement of the total ALDH activity and ALDH1A1 expression through the aryl hydrocarbon receptor (AhR) pathway [25]. Quercetin was also identified as an activator of the AhR pathway in the human hepatocyte model [28], suggesting the possible involvement of AhR in quercetin-induced cytoprotection. Since the supportive ALDH isozymes rather than ALDH2 might contribute to the cytoprotective effect of quercetin against acetaldehyde, quercetin is anticipated to prevent the adverse reactions associated with the alcohol intolerance caused by ALDH2 polymorphism, characteristic of East Asians.

The ROS overproduction linked to mitochondrial dysfunction might, at least in part, be attributed to the toxic mechanism of acetaldehyde [8]. Actually, we confirmed that the acetaldehyde treatment significantly increased the intracellular ROS level detected by the DCFH-DA assay (Figure 3). Similar to the cytotoxicity experiment, quercetin inhibited the intracellular ROS accumulation induced by acetaldehyde in a dose-dependent manner (Figure 3). These results raised the possibility that the inhibition of oxidative stress contributes to the cytoprotective effect of quercetin on acetaldehyde toxicity. To check this speculation, the gene expressions of representative antioxidant-related enzymes including HO-1, GCLC, and xCT were evaluated. As shown in Figure 4A, quercetin significantly up-regulated all the gene expressions. The role of HO-1 and its products, such as CO and biliverdin, in antioxidative cytoprotection against oxidative stress has been established in various experimental models including hepatocytes [11,29,30,31]. In addition to the function of HO-1, GSH acts as an intracellular antioxidant by directly scavenging ROS or serving as a substrate for glutathione peroxidases [32]. GSH up-regulation has also been suggested to be involved in the neuroprotection offered against hydrogen peroxide by quercetin [33]. Expectedly, quercetin also significantly increased the intracellular GSH level in Hepa1c1c7 cells (Figure 4B), which coincided with the enhanced gene expression related to not only GSH biosynthesis (GCLC), but also the substrate uptake (xCT) (Figure 4A). Taken together, quercetin might promote the production of endogenous antioxidant molecules.

In this study, we tried to clarify which endogenous antioxidant(s) play an important role in the inhibition of ROS accumulation by quercetin. The pharmacological experiments demonstrated that the inhibitor of HO-1 enzymatic activity, SNPP, significantly counteracted the inhibition of the acetaldehyde-induced ROS production by quercetin, whereas the effect of the inhibitor of glutamate-cysteine ligase, BSO, was very limited (Figure 5). These results implied that CO and biliverdin, rather than GSH, might be involved in the intracellular antioxidative action of quercetin. It should be noted that the microbiota catabolites of quercetin attenuated the acetaldehyde-induced oxidative stress dependent on the up-regulation of GSH and independent of HO-1 [22]. Furthermore, SNPP simultaneously abolished the cytoprotective effect of quercetin (Figure 6), suggesting that HO-1 and its antioxidative products might also contribute to the cytoprotection offered by quercetin. This suggestion is supported by a previous study showing that the HO-1-produced CO partly but significantly takes part in the inhibition of UVB-induced oxidative stress through the regulation of superoxide dismutase [34]. Bilirubin, a reduced product of biliverdin, has attracted attention as an effective agent for the remediation of oxidative stress caused by ischemia as well as hydrogen peroxide [35,36]. Taken together, quercetin protects hepatocytes from acetaldehyde-induced cytotoxicity, possibly by enhancing the intracellular antioxidant mechanism via HO-1-dependent CO/biliverdin production.

In our recent study, human hepatocyte models with ALDH isozyme deficiency were used to examine whether quercetin inhibits acetaldehyde toxicity [27]. Quercetin did not enhance the ALDH activity in ALDH1A1-deficient mutant (*aldh1a1-kd*), suggesting that ALDH1A1 mainly contributes the quercetin-enhanced ALDH activity but that ALDH2 does not. Nevertheless, quercetin significantly inhibited the acetaldehyde toxicity in the *aldh1a1-kd* cells, despite being weaker than the wild type [27]. The fact that quercetin inhibited the acetaldehyde-induced toxicity in the *aldh1a1-kd* cells without an enhancement in the ALDH activity suggested that other pathways besides ALDH-dependent mechanisms might be involved in quercetin-induced cytoprotection. In the present study, we identified that the HO-1-dependent pathway was the most likely the mechanism implicated in the quercetin-induced cytoprotection against acetaldehyde among the antioxidant-related mechanisms. Although it has been suggested that the endogenous antioxidant effects of quercetin are associated with the activation of the mitogen-activated protein kinases/Keap1/Nrf2 signaling pathway [37], the contribution of this pathway to the protective effects of quercetin in the experimental model of this study remains to be elucidated.

In conclusion, quercetin has the potential to prevent humans from acetaldehyde-dependent toxic events, including the inhibition of proliferation as well as intracellular ROS accumulation. The HO-1-dependent antioxidant production might be attributed to the cytoprotective effect of quercetin against acetaldehyde toxicity. Thus, the present study provides more evidence for the biological basis of quercetin as a health-promoting food chemical. On the other hand, the cultured cell model using the mouse hepatoma Hepa1c1c7 cells has certain limitations. The most important shortcoming is that this model does not reflect the characteristics of human normal hepatocytes. Next, the experimental model for the toxic events induced by acetaldehyde is temporary as well as acute. This model differs from the consecutive drinking-induced progression of chronic injury. In addition, quercetin itself has a toxic effect at higher concentrations, which reduces the cytoprotective effect at higher concentrations [31]. This results in a narrow effective concentration range of quercetin and difficulty observing a concentration dependence (Figure 1). On the other hand, quercetin concentration-dependently up-regulates not only the gene expression of ALDHs and antioxidant enzymes, but also ALDH activity and glutathione levels (Figure 2 and Figure 4). Therefore, we can only speculate that a possible reason for the cytoprotective effect lacking a clear concentration dependence is the presence of an independent toxic mechanism, but not a disturbance of the cytoprotective mechanism. Furthermore, the effective concentration of quercetin for the cytoprotection against acetaldehyde (~10 μM) might be far above physiological concentrations, since even the maximum plasma concentration of quercetin metabolites after the ingestion of a large number of onions (100 mg quercetin aglycone equivalent) was approximately 7.0 μM [38]. Therefore, future research is needed to clarify the efficacy of quercetin in in vivo rodent models and a human normal hepatic cell model. In addition to the details of the intracellular signaling pathways, it would be interesting to further understand how ALDHs and the antioxidant-producing enzymes differentially contribute to the cytoprotective effects of quercetin. Nevertheless, since quercetin enhanced acetaldehyde tolerance via the HO-1-dependent antioxidant mechanism, quercetin may improve the symptoms caused by ALDH2 polymorphism-dependent alcohol intolerance.

## 4. Materials and Methods

### 4.1. Materials

All chemicals except those listed below were obtained from FUJIFILM Wako Pure Chemical Corporation (Osaka, Japan) or Nacalai Tesque (Kyoto, Japan). α-Minimum essential medium (α-MEM) and TRIzol reagent; Life Technologies (Carlsbad, CA, USA). Fetal bovine serum (FBS); Nichirei Corporation (Tokyo, Japan). Quercetin dihydrate and L-buthionine-sulfoximine (BSO); Sigma Aldrich (St. Louis, MO, USA). ReverTra Ace; TOYOBO Co., Ltd. (Osaka, Japan). Taq polymerase; Takara Bio, Inc. (Kusatsu, Japan). The Bio-Rad Protein Assay; Bio-Rad Laboratories (Hercules, CA, USA). PierceTM BCA Protein Assay Kit; Scientific (Meridian Rd., Rockford, IL, USA). β-Nicotinamide-adenine dinucleotide, oxidized form (NAD^+^); Oriental Yeast Co., Ltd. (Tokyo, Japan). The anti-HO-1 antibody; Cell Signaling Technology (Danvers, MA, USA). Tin protoporphyrin (SNPP); Funakoshi Co., Ltd. (Tokyo, Japan).

### 4.2. Cell Cultures and Viability Determination

The mouse hepatoma cell line Hepa1c1c7, obtained from the American Type Culture Collection, was maintained as previously reported [22]. After the overnight preincubation of Hepa1c1c7 cells (1 × 10^4^ cells per well in a 96-well plate), quercetin was treated for 24 h. After the 3 h treatment with acetaldehyde (10 mM) for 3 h, the MTT assay was carried out as previously reported [22]

### 4.3. Intracellular Reactive Oxygen Species Measurement and Image Analysis

The intracellular ROS level was determined using the dichlorofluorescin diacetate (DCFH-DA) assay, as previously reported [22]. For the pharmacological experiments, SNPP or BSO was pretreated with the cells for 1 h. A Tali™ image-based cytometer was used to detect the intracellular fluorescence (Life Technologies).

### 4.4. RNA Extraction and Reverse Transcription-Polymerase Chain Reaction (RT-PCR)

Total RNA isolation, reverse transcription and PCR amplification were carried out as previously reported [22]. The primer sets are listed in Table 1.

### 4.5. Glutathione Titration

Glutathione (GSH) quantification was performed as previously reported [39].

### 4.6. ALDH Activity Assay

The ALDH activity was measured as previously described [25].

### 4.7. Western Blot Analysis

The protein expression of HO-1 was determined by western blotting as previously reported with some modifications [21]. After the 24 h treatment with quercetin, the cell lysates were obtained. An anti-HO-1 antibody was treated with the protein-transferred membrane at room temperature for 2 h. After the 1 h treatment of the secondary antibody conjugated with horseradish peroxidase, Chemi-Lumi One detection reagent was incubated for visualization. LAS-500 (Cytiva) was used for capturing the images.

### 4.8. Statistical Analysis

All the values represented as the mean ± SD were obtained from at least three different experiments. The Student’s paired two-tailed *t*-test or one-way analysis of variance (ANOVA) followed by Tukey’s honestly significant difference (HSD) test using SPSS 26.0 software (IBM, Chicago, IL, USA) were used to determine the statistical significance. A *p*-value less than 0.05 for all comparisons was considered significant.

## Figures and Tables

**Figure 1 ijms-25-09038-f001:**
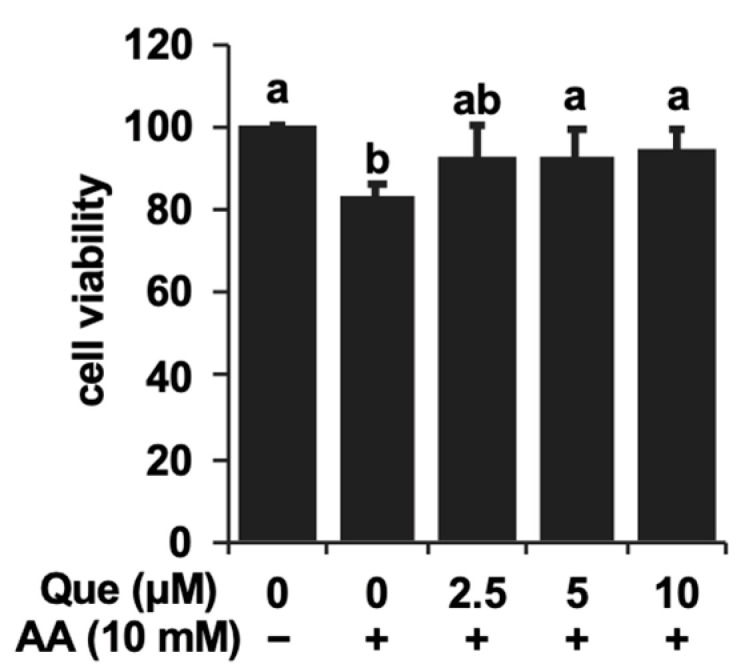
Quercetin protected Hepa1c1c7 cells from the cytotoxicity induced by acetaldehyde. After the 24 h pretreatment with quercetin (Que), acetaldehyde (AA, 10 mM) was treated with the cells for 3 h, then an MTT assay was carried out. All values are presented as the means ± SD of three separate experiments and subjected to one-way ANOVA using SPSS 26.0 software, followed by Tukey’s HSD. The different letters over the bars correspond to significant differences between treatments in each condition (*p* < 0.05).

**Figure 2 ijms-25-09038-f002:**
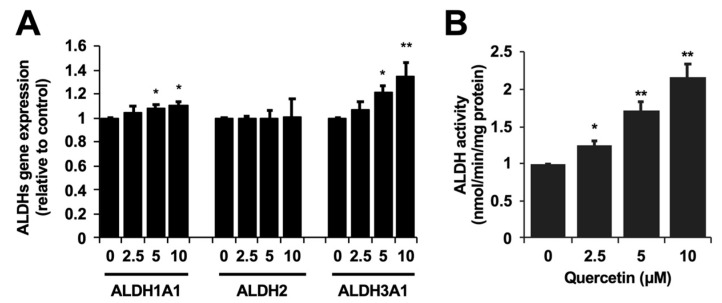
Quercetin modulated the ALDH gene expression and the ALDH activity in Hepa1c1c7 cells. (**A**) After the 6 h treatment with quercetin (Que), the total RNA was subjected to an RT-PCR analysis for each gene. (**B**) After the 24 h treatment with quercetin, the ALDH activity was evaluated. All values are presented as the means ± SD of three separate experiments and subjected to a Student’s *t*-test by using SPSS 26.0 software (**, *p* < 0.01; *, *p* < 0.05 vs. control).

**Figure 3 ijms-25-09038-f003:**
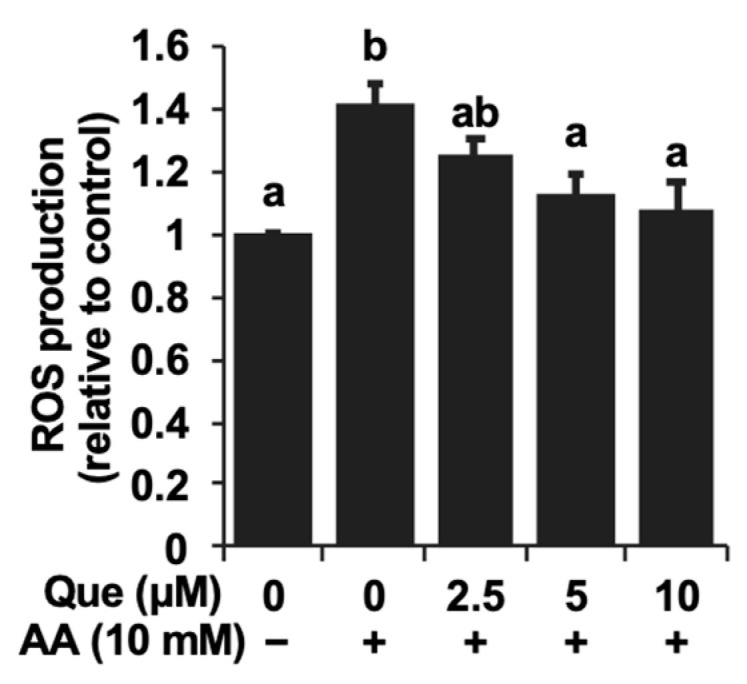
Quercetin inhibited the intracellular reactive oxygen species (ROS) accumulation induced by acetaldehyde. After the 24 h pretreatment with quercetin (Que), 10 mM of acetaldehyde (AA) was treated with the cells for 3 h. The cells were subjected to a dichlorofluorescin diacetate (DCFH-DA) assay using an image-based cytometer. All values are presented as the means ± SD of three separate experiments and subjected to one-way ANOVA using SPSS 26.0 software, followed by Tukey’s HSD. The different letters over the bars correspond to significant differences between treatments in each condition (*p* < 0.05).

**Figure 4 ijms-25-09038-f004:**
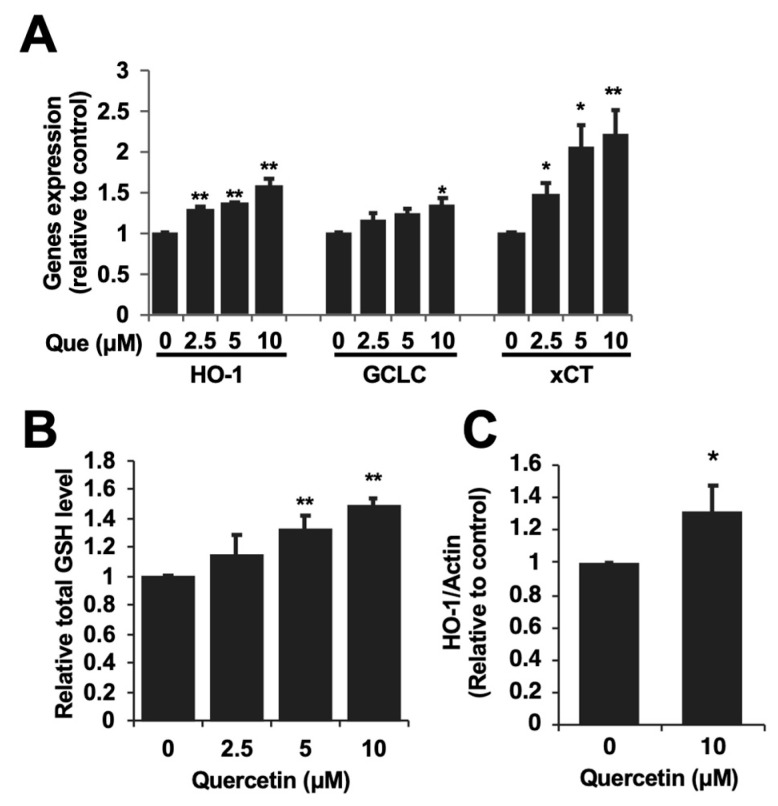
Quercetin enhanced the expression of the antioxidant-producing enzymes and the total glutathione level. (**A**) After the 6 h treatment with quercetin (Que), the total RNA was subjected to an RT-PCR analysis for each gene. (**B**) After the 24 h treatment with quercetin, the intracellular GSH level was evaluated. (**C**) After the 24 h treatment with quercetin (10 μM), the protein expression level of HO-1 was determined by western blotting. All values are presented as the means ± SD of three separate experiments and subjected to a Student’s *t*-test by using SPSS 26.0 software (**, *p* < 0.01; *, *p* < 0.05 vs. control).

**Figure 5 ijms-25-09038-f005:**
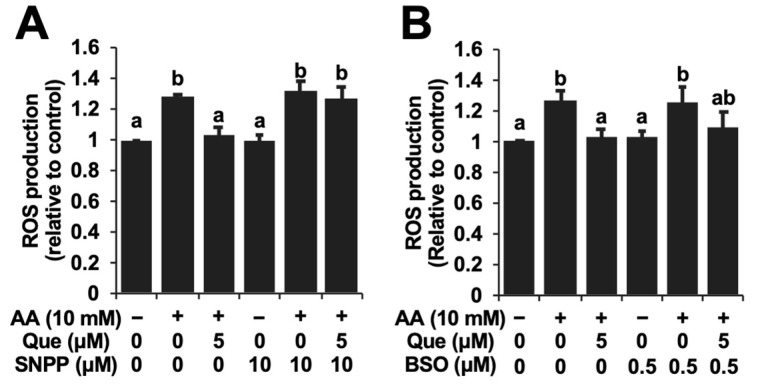
The HO-1 inhibitor, tin protoporphyrin IX (SNPP) (**A**), and the glutathione biosynthesis inhibitor, buthionine sulphoximine (BSO) (**B**), modulated the quercetin-induced inhibition of ROS accumulation. After the 1 h pretreatment with SNPP (10 μM) or BSO (0.5 μM), quercetin (Que) was treated with the cells for 24 h. After the 3 h treatment with 10 mM of acetaldehyde (AA), the cells were subjected to the DCFH-DA assay. All values are presented as the means ± SD of three separate experiments and subjected to one-way ANOVA using SPSS 26.0 software, followed by Tukey’s HSD. The different letters over the bars correspond to significant differences between treatments in each condition (*p* < 0.05).

**Figure 6 ijms-25-09038-f006:**
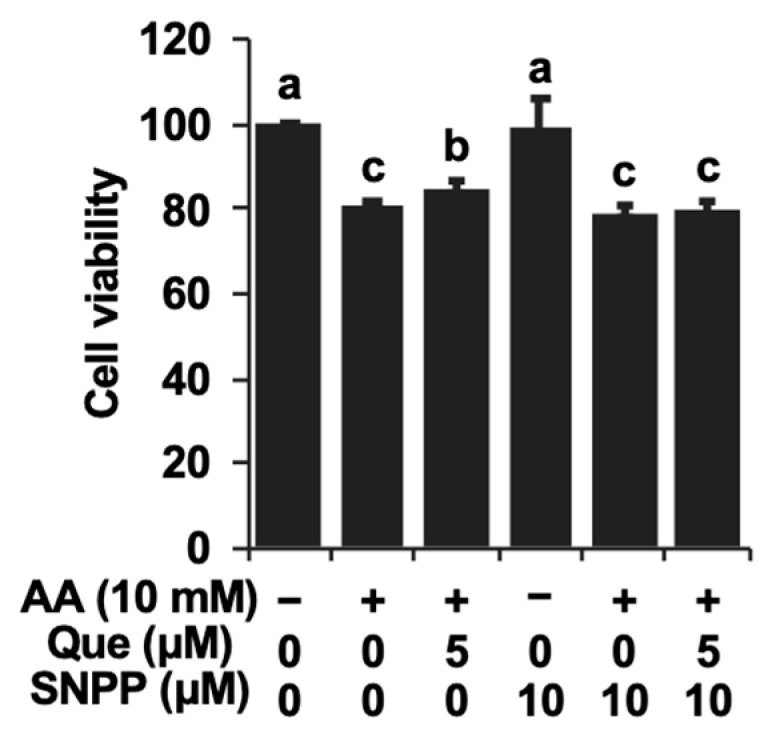
The HO-1 inhibitor cancelled the quercetin-induced cytoprotection against acetaldehyde. After the 1 h pretreatment with SNPP (10 μM), quercetin was treated with the cells for 24 h. After the 3 h treatment with 10 mM of acetaldehyde, the cells were subjected to an MTT assay. All values are presented as the means ± SD of three separate experiments and subjected to one-way ANOVA using SPSS 26.0 software, followed by Tukey’s HSD. The different letters over the bars correspond to significant differences between treatments in each condition (*p* < 0.05).

**Table 1 ijms-25-09038-t001:** Primer sets for this study.

Gene	Forward Primer	Reverse Primer	Cycles and Product Size
*mβ-actin*	5′-GTCACCCACACTGTGCCCATCTA-3′	5′-GCAATGCCAGGGTACATGGTGGT-3′	16, 455 bp
*mALDH1A1*	5′-GACAGGCTTTCCAGATTGGCTC-3′	5′-AAGACTTTCCCACCATTGAGTGC-3′	26, 142 bp
*mALDH2*	5′-TGAAGACGGTTACTGTCAAAGTGC-3′	5′-AGTGTGTGTGGCGGTTTTTCTC-3′	26, 115 bp
*mALDH3*	5′-GATGCCCATTGTGTGTGTTCG-3′	5′-CCACCGCTTGATGTCTCTGC-3′	26, 138 bp
*mHO-1*	5′-ACATCGACAGCCCCACCAAGTTCAA-3′	5′-CTGACGAAGTGACGCCATCTGTGAG-3′	22, 203 bp
*mGCLC*	5′-GGCGATGTTCTTGAGACTCTGC-3′	5′-TTCCTTCGATCATGTAACTCCCATA-3′	26, 100 bp
*mxCT*	5′-CCTGGCATTTGGACGCTACAT-3′	5′-TGAGAATTGCTGTGAGCTTGCA-3′	25, 182 bp

## Data Availability

The original contributions presented in the study are included in the article. Further inquiries can be directed to the corresponding author.

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
