# Peer review of "Quercetin Attenuates Acetaldehyde-Induced Cytotoxicity via the Heme Oxygenase-1-Dependent Antioxidant Mechanism in Hepatocytes"

_ijms, 2024, doi:10.3390/ijms25169038_

Round 1

Reviewer 1 Report

Comments and Suggestions for Authors

The major findings of the manuscript is described that pre-treatment of quercetin attenuated the cytotoxicity induced by acetaldehyde by upregulating antioxidative pathways including HO-1, GCLC, xCT, and GSH level. All theses findings are derived from in vitro experiment using mouse hepatoma Hepa1c1c7 cells. The manuscript must address following major issues.

1.  Hepa-1 was derived from the BW7756 hepatoma that developed in a C57L mouse. Hepa-1c1c7 cells express the aryl hydrocarbon (Ah) receptor and were shown to be highly inducible for cytochrome P450IA1. Increased level of AhR has been shown to inhibit cytotoxicity of acetaldehyde in previous studies. Moreover, the metabolic pathways of P450IA1 is involved in acetaldehyde cytotoxicity. The results shown in the manuscript should be reproduced in other liver cell line or in vivo study.

2. Cytotoxicity of acetaldehyde in this cell line must be established first by showing dose-dependent cytotoxicity. In addition, in Fig 1, cytoprotective effects of quercetin against acetaldehyde appear not to be concentration dependent.

3. In numerous previous studies, quercetin has been known as a potent inhibitor of the dehydrogenase action of cytosolic aldehyde dehydrogenase over most of the concentration range. The result of the manuscript must address the previous observations and provide more results including pathway analysis.

4. Up-regulation of ALDH, HO-1, GCLC, and cCT by quercetin was all determined only by determining the level of gene expression. The change of the enzymatic activity as well as protein level must be shown.

Comments on the Quality of English Language

Minor editing of English language required

Reviewer 2 Report

Comments and Suggestions for Authors

This paper is an interesting contribution to the use of quercetin to enhance acetaldehyde-induced resistance cytotoxicity in mouse hepatoma Hepa1c1c7 cells. The experiment is well designed and carried out and the conclusions are well supported by the results. The limitations of the study are clearly described as is future work. The paper is well written and is largely error-free. My only query is on L 328 - is the absorbance change measured after 30 min (rather than over 30 min)?

Reviewer 3 Report

Comments and Suggestions for Authors

This paper by Li et al. evaluated quercetin as a potential cytoprotective against the acetaldehyde-induced cytotoxicity in the cultured hepatocyte model. Their main focus was on ALDH isozymes and how this may alter after acetaldehyde exposure and quercetin treatment.

1-This manuscript's title should be rephrased because the word resistance is misleading. consider options like "protective effects" or "modulation". The introduction was comprehensive and the relevant references were cited, however, they selected to test quercetin's antioxidant and cytoprotective effects on Hepa1c1c7 cells are well-established so the novelty must be highlighted further in the abstract and introduction.

2-In the result section, lines 90-91, revise the following statement “After the 24-h pretreatment of quercetin, acetaldehyde (10 mM) was treated with Hepa1c1c7 cells for 3 h.”

Also, in the Figure 1 caption, the authors used XLSTAT. While in methods they mentioned only SPSS. if you used both SPSS and XLSTAT, specify which analysis used which software.

3-In the figure captions, the authors repeatedly used this statement ‘The different letters above the bars indicate significant differences among the treatments for each condition (p < 0.05).” Using letters in most of the figures such as a, ab, b and not clear to me, add asterisks instead and indicate the p-value is less than 0.05 between which groups exactly.

4-In Figure 2, the cells are only treated with quercetin and the gene expressions of ALDH isozymes and enzyme activities are quantified, also the same issue was found in Figure 4A, these figures should be repeated in the presence of acetaldehyde to understand its interaction with quercetin.

The discussion and conclusion were parallel with the obtained results.

5-In the PCR methodology, it is better if the primers are listed in a table.

6-The authors used several concentrations of quercetin based on what? Can these doses be used physiologically?

7- For statistical analysis, they used a paired t-test what are the reasons for using this test not the unpaired test? I think you don’t compare the same cells pre-treatment and post-treatment to use paired t-tests.

Round 2

Reviewer 1 Report

Comments and Suggestions for Authors

Although the authors provided the responses for the reviewer's concerns, any of the major issues was properly addressed. 

As the authors noted, if the findings presented in their recently published paper, "Evaluation of quercetin as a potential cytoprotector against acetaldehyde using the cultured hepatocyte model with aldehyde dehydrogenase isozyme deficiency", demonstrated the effect of quercetin on acetaldehyde-induced Cy- 2 cytotoxicity in hepatocytes, the significance of the new findings in this manuscript seems less significant. In addition, the study of protein changes and signaling mechanisms for the mechanistic study of the effects mediated by HO-1 is insufficient.

The authors explained that quercetin itself has a toxic effect at higher concentrations, and thus the cytoprotective effect decreases at higher concentrations. The authors also pointed out that the effective concentration range of quercetin is narrow and it is difficult to observe the concentration dependence. However, these results and explanations are precisely what this paper means by pointing out that the effect of quercetin is very limited and therefore not universal. It is a counter-argument that more detailed mechanistic studies should be conducted to prove the effect of quercetin. In addition, the authors argued that they have mentioned this point in the discussion section with a proper reference. However, the references provided by the authors not only do not show results on the toxicity of quercetin, but also show that quercetin 4′-O-β-glucoside is the major metabolite of quercetin, so this also suggests that the effects of quercetin 4′-O-β-glucoside should have been tested in parallel with quercetin in this manuscript.

In the Western blot results presented in Supplemental Fig. 4A, the beta-actin, which is the standard for quantification, and HO-1, GCLC, and xCT, which are actually to be quantified, were analyzed in different lanes, indicating that the quantification of the corresponding proteins was not performed properly. In particular, in some blots, different gel images were stitched together, which raises concerns about the distortion of the experimental results.

Reviewer 3 Report

Comments and Suggestions for Authors

The modifications are clear 

Author Response

The modifications are clear.

The comments offered by the reviewers have been helpful in formulating what we believe is a stronger paper.  We deeply appreciate these thoughtful comments.

Round 3

Reviewer 1 Report

Comments and Suggestions for Authors

This manuscript elucidates the antioxidant mechanism of quercetin, particularly dependent on heme oxygenase-1, in its effect of alleviating acetaldehyde toxicity. In addition to the previously known protective effect of quercetin against alcohol and acetaldehyde toxicity, it is meaningful to elucidate the heme oxygenase-1-dependent mechanism. However, the experimental method for elucidating the mechanism in cell-level experiments is limited and lacks scalability, so it is regrettable that the results presented in the paper alone cannot fully support the conclusions suggested by the authors. Nevertheless, the results described in the paper clearly present scientific evidences, and thus, in addition to the results proven in previous papers, a new mechanism of quercetin that acts to alleviate acetaldehyde toxicity is well described.

Author Response

Thank you very much for your insightful comment, which have made our manuscript much better than before.  We are also very pleased that you have given understanding to our responses.  We are also very pleased with your consideration and understanding of our response, even though our results alone are somewhat insufficient to support our conclusions.  Thank you again for your kind consideration.